# MedSQLX: Translation of Medical Queries into UDF-Centric SQL

Catlynh Nguyen
cnguye98@asu.edu
Arizona State University
Tempe, Arizona, United States

Irbaz Bin Riaz
riaz.irbaz@mayo.edu
Mayo Clinic - Arizona
Scottsdale, Arizona, United States

Chitta Baral
cbaral@asu.edu
Arizona State University
Tempe, Arizona, United States

Jia Zou
jia.zou@asu.edu
Arizona State University
Tempe, Arizona, United States

## ABSTRACT

Natural Language to Structured Query Language (NL-to-SQL) systems have matured considerably for standard SQL generation, yet they largely overlook user-defined functions (UDFs), which are custom higher-order logic, such as clinical rules, scoring formulas, and domain-specific classifications.

This paper presents an agentic framework that leverages Large Language Models (LLMs) for generating UDF-centric queries from natural language task descriptions in the medical domain. Our proposed framework has two layers: a semantic analysis layer and a code generation layer. The semantic analysis layer consists of one detector agent that identifies ambiguity of the task description, and one resolver agent that ranks potential interpretations for each ambiguous aspect of the query. The code generation layer is built on top of the AgentFlow paradigm, where a planner agent coordinates three specialized tools: a medical knowledge base query executor, a DuckDB query executor, and an LLM-based UDF validator to iteratively generate, register, and verify DuckDB SQL macro functions and Python UDFs. Once the verifier agent decides that the task has been completed, a generator agent will generate the complete SQL code with UDF invocations.

We evaluate the system on ten medical query scenarios using the MedAgentBench dataset, including glucose classification, potassium risk, chronic kidney disease staging, HbA1C classification, creatinine elevation, temporal recency, multi-factor diabetes risk scoring, glucose trend detection, creatinine streak analysis, and potassium volatility assessment. Our framework correctly handles all 57 test cases with 100% accuracy, compared to 49% accuracy achieved by DIN-SQL, a state-of-the-art NL-to-SQL baseline, demonstrating that structured tool orchestration with verification loops substantially improves generation quality.

**VLDB Workshop Reference Format:**
Catlynh Nguyen, Irbaz Bin Riaz, Chitta Baral, and Jia Zou. MedSQLX: Translation of Medical Queries into UDF-Centric SQL. VLDB 2026 Workshop: Biomedical Data Management Systems (BioDMS).

## 1 INTRODUCTION

Translating natural language (NL) into SQL has received significant attention recently. Various LLM-driven NL-to-SQL paradigms have emerged, including zero-shot/few-shot learning techniques [7, 17, 18] and finetuning-based techniques [13, 19, 25]. In the medical domain, structured query generation over clinical databases is particularly valuable. Clinicians, nurses, and researchers need to extract insights from electronic health records without requiring deep SQL expertise [8, 12]. However, most of the existing NL-to-SQL systems assume that all required processing logic on top of relational data can be represented as SQL. In medical and clinical analytics on relational data, this assumption frequently breaks down. Clinicians and data analysts routinely need functions that encode domain logic, such as classifying a glucose reading as diabetic, staging chronic kidney disease from an Estimated Glomerular Filtration Rate (eGFR) value, flagging creatinine elevation based on patient gender, temporal streak detection, and multi-factor risk scoring. These computations require user-defined functions (UDFs) that encode complicated clinical domain knowledge.

A natural approach is to adapt recent advances in code generation, which leverage large language models to synthesize executable programs from natural language specifications, particularly in general-purpose languages such as Python. Early systems such as Codex, along with more recent models including GPT-4 and Code Llama, demonstrate strong performance on benchmarks such as HumanEval [3] and MBPP [1], which evaluate functional correctness using unit tests [2, 4, 21]. Open models such as StarCoder, DeepSeek-Coder, and CodeT5+ further expand accessibility and support multi-language code generation. Subsequent work extends this paradigm to multilingual settings and broader domains [15, 16, 20], while interactive approaches incorporate test-driven refinement loops to improve alignment with user intent [6, 11].

Despite their success, these approaches cannot be directly applied to generate SQL queries with UDFs. First, they primarily target standalone program synthesis, where the output is a self-contained function, whereas UDF-centric query generation requires tight integration between generated functions and relational queries, including correct invocation, data typing, and interaction with database execution semantics. Second, these methods typically assume that the natural language specification is complete and unambiguous, while real-world tasks, especially in the medical domain, often involve underspecified or conflicting requirements that must be resolved before code generation. Third, correctness in prior work

is usually validated via unit tests or surface-level execution, which is insufficient for UDFs that encode complex domain logic; ensuring correctness requires data-aware and domain-aware validation (e.g., clinical consistency and distributional sanity checks). Fourth, existing approaches lack iterative, tool-augmented workflows that jointly reason over knowledge bases, data systems, and generated code, which are essential for constructing, registering, and verifying UDFs within SQL pipelines. As a result, directly applying these techniques often leads to syntactically valid but semantically incorrect or non-executable SQL+UDF programs.

**MEDSQLX: Translating Medical Queries to UDF-Centric SQL.** To close these gaps, we propose a novel LLM-driven agentic framework to automatically translate medical queries to UDF-centric SQL. To combine the clinical domain knowledge, the UDF generation, and the generation of the end-to-end SQL queries that nest with these UDFs, we adopt a two-layer architecture, with the top layer analyzing the semantics of the natural language query to infer the intention of the query, and the bottom layer generating the UDF-centric SQL query.

At the top layer, we proposed a chain of clarifications (COC) reasoning process, where a detector agent discovers all underspecified phrases, and for each such phrase, a resolver agent uses LLM or tools to provide a most probable clarification, which is used to augment the context.

The augmented contexts are passed to the bottom layer, where the UDFs and the final UDF-centric SQL query is generated in a self-correcting loop that generates, registers, and validates UDFs against a live medical database. At this layer, we adopt an AgentFlow-style reasoning process [14] that consists of four agents: planner, executor, verifier and generator. In each iteration, (1) the planner node analyzes the contexts and agent interaction memory in past iterations, and chooses a tool to execute, (2) the executor node executes the tool, and (3) the verifier node decides whether the task has completed based on the tool output and the historical interaction memory. These steps will repeat until the verifier considers the task as completed. Then, the generator node will generate and provide the final response, the end-to-end UDF-centric SQL query, to the user. Our current implementation incorporates three tools for retrieving medical domain knowledge, executing generated UDFs with DuckDB, and validating generated UDFs, respectively.

**The contributions of this work** are summarized as follows:
• We are the first to identify and address the problem of translating clinical questions in natural language into UDF-centric SQL queries on structured patient data.
• We proposed a novel two-layer agentic framework that combines a Chain of Clarifications reasoning strategy for query semantic analysis and AgentFlow for hybrid UDF/SQL code generation.
• We implemented a prototype of the proposed framework and memory compression techniques, and use our extended MedAgent-Bench [8] to evaluate it on 57 test cases from ten representative clinical query scenarios. The results showed that our framework significantly outperformed the state-of-the-art NL-to-SQL frameworks such as DIN-SQL by 2× in accuracy.

## 2 SYSTEM OVERVIEW

Our system consists of two layers, detailed in the following sections. The two-layer design reflects a principled separation of concerns. Semantic analysis and code generation have fundamentally different computational profiles and failure modes. Underspecification resolution is a reasoning task over natural language and domain knowledge, while code generation is an execution-grounded task requiring live database interaction and iterative validation. Resolving ambiguity before code generation prevents irreversible commitment to incorrect interpretations mid-generation. Once an agent has registered a UDF under a wrong assumption, recovery requires restarting rather than revising. Separating the two layers also improves modularity. Each layer can be developed, evaluated, and replaced independently. For instance, the Chain of Clarifications resolver in the Semantic Analysis Layer could be replaced with an alternative resolution strategy without modifying the Agent-Flow code generation layer, and vice versa. Figure 1 shows the full pipeline from natural language query to UDF-centric SQL output.

### 2.1 Semantic Analysis Layer

This layer addresses a core limitation: natural language medical queries are frequently underspecified with respect to UDF generation. A question like *"What is the longest consecutive streak of elevated creatinine readings for this patient?"* leaves multiple aspects unstated: (1) the definition of elevated creatinine, which should provide *gender/age-specific* clinical thresholds, (2) the output type, which should be an integer count of consecutive readings, inferred from the phrasing "longest consecutive streak", (3) the input format, which should be a list of readings with timestamps, inferred from the need to assess ordering and consecutiveness, and (4) the time frame for what constitutes a streak.

Existing approaches to underspecification include asking a human for clarification [5, 23] and retrieving knowledge from pre-existing documents [22, 24, 26]. Our pipeline resolves underspecification more autonomously, leveraging two LLM agents, using a reasoning strategy, which is termed as **Chain of Clarifications**.

*2.1.1 Underspecification Detector.* The detector takes the natural language question and identifies a list of ambiguous or unclear aspects to be clarified. It further categorizes each gap into one of three: gaps that existing tools can handle autonomously (tool-handled), gaps that can be probabilistically resolved from question phrasing (resolvable), and gaps that are unresolvable.

*2.1.2 Underspecification Resolver.* For tool-handled gaps, the resolver records that the downstream tool is expected to resolve the missing information and proceeds without further disambiguation. For resolvable gaps, the resolver generates multiple candidate interpretations and ranks them using heuristic plausibility signals derived from the query phrasing and available tool metadata. Since the resolver relies on the same underlying LLM as the rest of the pipeline, the ranking reflects model-internal preferences rather than an independently validated measure of uncertainty. The highest-ranked interpretation is selected for downstream planning, while alternative candidates may optionally be retained for fallback or verification. For unresolvable gaps, the resolver explicitly flags the ambiguity and instructs the planner in the code generation layer to

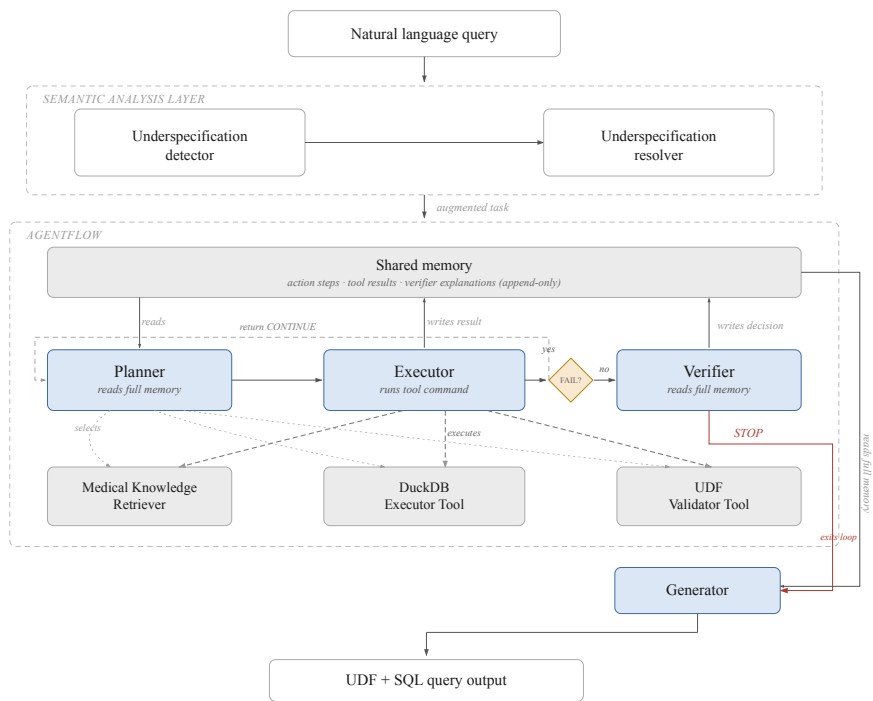

**Figure 1: MedSQLX system architecture. The Semantic Analysis Layer resolves underspecification in the natural language query before passing an augmented task to the AgentFlow code generation layer. The Planner selects tools, the Executor runs them, and the Verifier checks completion after each step. On STOP, the Generator reads full shared memory and produces the final UDF and SQL query.**

make an implementation decision and document the assumption in the generated workflow.

The resolver generates an assumption block prepended to the task prompt, as shown in Fig. 2.

```
CONTEXT AUGMENTATION: These assumptions were
    derived from the question phrasing. They are
    not ground truth - treat them as informed
    priors.
- definition of volatility: assume frequent
    fluctuations measured by standard deviation
  Basis: 'unusually volatile' suggests magnitude
     of variation rather than frequency of
     direction changes.
- output type: assume VARCHAR categorical label
  Basis: 'are levels unusually volatile' implies a
      yes/no or categorical answer rather than a
     numeric value.

The following gaps could not be resolved - use
    your best judgment and document the choice
    explicitly:
- time frame for volatility assessment
```

**Figure 2: Block generated by the underspecification resolver for the query *"Are this patient's potassium levels unusually volatile?"***

## 2.2 Code Generation Layer

The code generation layer is built on top of AgentFlow [14], an agentic framework designed for improved reasoning and tool-augmented tasks by coordinating planner, executor, verifier, and generator agents through an evolving, shared memory, iterative process.

*2.2.1 Planner.* The planner handles two functions at each step: initial query analysis and next-step prediction. At the start of a run, `analyze_query()` receives the task prompt and available tool metadata and produces a concise structured analysis identifying the main objectives, relevant tools, and additional considerations. This analysis is passed as context to every subsequent planning step. At each step, `generate_next_step()` receives the query, the query analysis, the full memory of previous actions, and current step count. It selects a single tool and formulates a sub-goal with all necessary context for that tool to execute.

*2.2.2 Tools.* In the current implementation, we design and exploit the following tools, which can be easily extended to improve the capability of the framework.

**Medical Knowledge Retriever** acts as a proxy for a clinical knowledge base API. For example, it accepts a lab test name (e.g., glucose) and returns clinically validated threshold ranges, domain context, sentinel bounds for values outside the clinically valid measurement

range, and expected return types. The tool provides medical knowledge only and does not include implementation instructions or code-generation guidance, leaving all design decisions to the agent.

**DuckDB Executor Tool** handles all database interactions: schema inspection, SQL macro UDF registration via `CREATE OR REPLACE MACRO`, Python UDF registration via DuckDB's `create_function` API, test execution, and macro cleanup for retries. A pre-registration check detects typed macro parameters and returns a corrective error message with the untyped form, allowing the agent to fix and retry without manual intervention. For Python UDFs, the tool handles registration state management across scenarios and converts failure responses into explicit error signals, enabling the agent to self-correct rather than loop indefinitely on unrecoverable states.

**UDF Validator Tool** uses LLM to perform two checks: type compliance (i.e., does the UDF return a type consistent with its declared signature?) and logic critique (i.e., is the UDF clinically sound and does it handle edge cases correctly?). The critique evaluates medical domain-specific threshold encoding, algorithmic correctness, edge case handling, and clinical appropriateness. When issues are found, the validator returns corrected UDFs and registers them.

*2.2.3 Verifier.* The verifier evaluates memory after each tool execution and decides whether the agent should continue or stop. It receives full memory view and returns a structured STOP or CONTINUE decision with a natural language explanation. The prompt specifies five completion criteria: threshold retrieval, schema inspection, UDF registration with success status, type compliance passing, and logic critique runs successfully.

To address a reliability issue in which the verifier occasionally returned STOP when the last memory step contained an explicit FAIL verdict from UDF Validator Tool, we added a deterministic check before the LLM call. If the most recent action step contains a FAIL verdict, the verifier immediately returns CONTINUE without making any API call. This fix eliminates false-positive stopping decisions at zero token cost.

*2.2.4 Generator.* After the verifier returns STOP, the generator produces two outputs. The first is a detailed, structured breakdown of the full process. For each step it describes the tool used, its purpose, and the key result obtained, then synthesizes a conclusion. This output is intended as a human-readable audit trail of the agent's reasoning. The second is a concise direct response to the original query, synthesizing key findings into a step-by-step summary and providing a precise answer.

## 2.3 End-to-End Example

We illustrate the full MedSQLX pipeline on the potassium volatility scenario. Given the query *"Are this patient's potassium levels unusually volatile?"*, the **Underspecification Detector** identifies three ambiguous aspects: (1) the *definition of volatility* - what computational measure defines "unusually volatile"; (2) the *output type* - whether the answer should be a boolean, numeric score, or categorical label; and (3) the *time frame* - what window of readings to consider. The **Underspecification Resolver** categorizes the first two as resolveable and the third as unresolvable, producing the assumption block shown in Figure 2.

The augmented task description is passed to the Code Generation Layer. Through an iterative planning loop, the Planner selects tools in sequence: it first calls the *Medical Knowledge Retriever* with `lab_test=potassium`, retrieving the normal reference range and sentinel bounds for physiologically invalid readings. It then calls the *DuckDB Executor Tool* to inspect the schema, confirming that potassium readings are stored as VARCHAR values under observation code K in the `observations` table. The Executor runs each tool call and returns results to shared memory. The Verifier evaluates memory after each step, checking that threshold retrieval, schema inspection, UDF registration, type compliance, and logic critique have all completed successfully before returning STOP.

Once the Verifier returns STOP, the Generator produces two outputs: a structured audit trail of the full agent reasoning process, and the final executable response, which includes both the registered Python UDF and the UDF-centric SQL query that invokes it:

```python
def potassium_volatility(readings):
    import numpy as np
    if not readings or len(readings) < 2:
        return 'Insufficient_data'
    values = [float(v) for v in readings
              if v is not None]
    if len(values) < 2:
        return 'Insufficient_data'
    volatility = np.std(values) / np.mean(values)
    return 'Volatile' if volatility > 0.1 else '
        Stable'
```

```sql
SELECT p.patient_id,
    potassium_volatility(
        LIST(o.value ORDER BY o.effectiveDateTime)
    ) AS volatility_status
FROM patients p
JOIN observations o
    ON p.patient_id = o.patient_id
WHERE o.code = 'K'
GROUP BY p.patient_id;
```

The UDF Validator Tool confirmed type compliance and clinical soundness prior to final generation, noting that coefficient of variation appropriately captures electrolyte stability and that edge cases including empty input and single-reading sequences are handled correctly.

## 3 PRELIMINARY EVALUATION

### 3.1 Evaluation Setup

We use OpenAI's GPT-4o API to implement all LLM agents. We use the MedAgentBench database [8], which contains 785,207 medical records derived from Stanford Hospital's STARR dataset, via a Fast Healthcare Interoperability Resources (FHIR) server and loaded into DuckDB as relational tables. FHIR is the standard for health data exchange used by modern hospital systems, so this structure mirrors what a real electronic health record (EHR) would expose to a query interface. Our UDFs operate over an `observations` table containing longitudinal lab results with fields for observation code, value, unit, and timestamp, and a `patients` table with demographics including gender and birthdate. Lab values in the `observations` table are stored as VARCHAR strings rather than numeric types,

**Table 1: All ten evaluation scenarios with natural language queries.**

| Name | Natural Language Query | UDF Required |
|---|---|---|
| glucose | Show me patients with high glucose levels indicating diabetes risk | classify_glucose(value) |
| potassium | Find patients with dangerous potassium levels | classify_potassium(value) |
| CKD Staging | Show patients grouped by chronic kidney disease stage | classify_ckd_stage(value) |
| HbA1C | Show me which patients are diabetic based on their HbA1C levels | classify_hba1c(value) |
| creatinine | Find patients with elevated creatinine levels considering gender differences | classify_creatinine(value, gender) |
| days since | For each patient, calculate how many days since their last glucose test | days_since(test_date) |
| diabetes risk | Calculate a diabetes risk score for each patient based on their age, glucose, and HbA1C | calculate_diabetes_risk(age, glucose, hba1c) |
| glucose trend | Is this patient's glucose getting worse over time? | glucose_trend(readings) |
| creatinine streak | What is the longest consecutive streak of elevated creatinine readings for this patient? | longest_elevated_streak(readings, gender) |
| potassium vol | Are this patient's potassium levels unusually volatile? | potassium_volatility(readings) |

reflecting common EHR export formats where values may include qualitative indicators alongside numeric readings.

**Query Scenarios** We designed ten query scenarios that rely on UDFs encoding domain knowledge. These UDFs cover classification, boolean detection, temporal arithmetic, multi-factor scoring, glucose trend detection, creatinine consecutive streak analysis, and potassium volatility assessment. Each scenario involves a UDF-centric SQL query generation task defined by a natural language question, as shown in Table 1. Each scenario has multiple test cases covering different regions of the input space, including normal values, boundary conditions, edge cases, and sentinel values. A test case is an input-output pair for one of the scenarios. In total, 57 test cases are distributed across these scenarios.

**Baseline** We use DIN-SQL [18], the state-of-the-art NL-to-SQL tool, as the baseline for evaluation. DIN-SQL was selected because it supports the same underlying model (GPT-4o), isolating the contribution of tool orchestration and verification from model choice, and because it represents a strong prompting-based approach against which the value of specialized tooling can be measured.

**Evaluation Metric.** We evaluate MedSQLX against DIN-SQL across all 57 test cases in ten scenarios using a unified evaluation framework. UDFs with deterministic resolvable underspecifications (e.g., in the first seven scenarios) use exact-match ground truth test cases written manually against clinically validated thresholds. UDFs with unresolvable/non-deterministic underspecifications (e.g., in the last three scenarios) use behavioral ground truth test cases that assert correct clinical interpretation without specifying output vocabulary, accepting valid design variation in implementation choices.

## 3.2 Results and Analysis

MedSQLX achieves 57 of 57 test cases (100%), compared to 28 of 57 (49%) for the DIN-SQL baseline overall. On scenarios involving resolvable underspecifications, MedSQLX achieves 39/39 (100%) versus DIN-SQL's 28/39 (72%). Three scenarios where both systems succeed (glucose, potassium, CKD staging) involve straightforward single-parameter classification, with minimal domain knowledge gaps. The four scenarios where DIN-SQL fails each expose a structural limitation of prompting-only generation that tool orchestration directly addresses. For **creatinine**, DIN-SQL generates typed macro parameters that cause a DuckDB registration error with no recovery path, as it has no mechanism to detect the error type and generate a corrective response. In MedSQLX, the `DuckDB Executor Tool` pre-registration check detects this and returns a correct error

message, allowing the agent to fix and re-register successfully. For **HbA1C** and **days since**, DIN-SQL lacks access to domain knowledge. It applies a generic sentinel upper bound rather than the HbA1C-specific bound of 20, and uses a hardcoded reference date rather than `CURRENT_DATE`. In MedSQLX, the `Medical Knowledge Retriever` provides this domain knowledge directly, resolving both issues. For **diabetes risk**, the failure is not a knowledge gap but a reasoning failure. DIN-SQL's self-correction identifies the spurious age bracket but does not apply the fix, and hallucinates a weighted average formula instead of the point-based scoring table. In MedSQLX, the `UDF Validator Tool`'s deterministic check catches and rejects the spurious bracket, and the scoring table returned by `Medical Knowledge Retriever` gives the agent the correct specification to follow.

DIN-SQL failed on the last three scenarios due to architectural mismatch. DIN-SQL generates relational SQL queries. While the UDF logics in the first seven scenarios are nested threshold comparisons, which can be translated to extended SQL equivalents using CASE/WHEN structures, the last three scenarios require Python UDFs expressing sequence analysis and statistical computation that SQL cannot express natively. DIN-SQL produces syntactically valid SQL queries for these scenarios, but queries rather than callable functions. The generated SQL retrieves or aggregates data from the observations table, but cannot be registered as a scalar UDF invocable from SQL expressions, which is what these scenarios require.

**Table 2: Per-scenario correctness results.**

| Scenario | MedSQLX | DIN-SQL | Evaluation |
|---|---|---|---|
| glucose | 5/5 (100%) | 5/5 (100%) | exact match |
| potassium | 5/5 (100%) | 5/5 (100%) | exact match |
| CKD Staging | 6/6 (100%) | 6/6 (100%) | exact match |
| HbA1C | 6/6 (100%) | 3/6 (50%) | exact match |
| creatinine | 6/6 (100%) | 0/6 (0%) | exact match |
| days since | 6/6 (100%) | 0/6 (0%) | exact match |
| diabetes risk | 5/5 (100%) | 0/5 (0%) | exact match |
| glucose trend | 6/6 (100%) | 0/6 (0%) | behavioral |
| creatinine streak | 6/6 (100%) | 0/6 (0%) | behavioral |
| potassium vol. | 6/6 (100%) | 0/6 (0%) | behavioral |
| **Overall** | **57/57 (100%)** | **28/57 (49%)** | |

For the **glucose trend** scenario, MEDSQLX achieved 6/6 tests passing. It autonomously chose to return a categorical trend label (*Consistently rising*, *Consistently falling*, *Variable*), which was a design decision not specified anywhere in the scenario. For the **creatinine streak** scenario, in MEDSQLX, the validator returned `NEEDS_REVIEW` on an intermediate version, citing missing input validation for edge cases, and the agent registered a corrected final version which passed all test cases. Notably, the function signature evolved autonomously across runs from a numeric threshold parameter to a gender-based lookup, reflecting the agent's interpretation that creatinine thresholds differ by gender, which is a clinically correct inference from the question phrasing alone. In the **potassium volatility** scenario, MEDSQLX achieved 6/6 tests passing at high confidence. The planner agent chose standard deviation as its volatility measure and a categorical output label, inferred from the question phrasing *"Are this patient's potassium levels unusually volatile?"* with no specification provided for "unusually".

### 3.3 Limitations

**Evaluation benchmark scope.** The current evaluation uses 57 test cases distributed across ten manually-written scenarios, all derived from the same MedAgentBench database of 100 Stanford Hospital patients. While the test cases were fixed prior to final evaluation runs, the scenarios were designed in conjunction with system development, and the benchmark does not yet include a separate held-out test set. The 100% pass rate demonstrates that structured tool orchestration with verification loops can correctly handle the range of scenario types in our benchmark, including scenarios that expose structural limitations in prompting-only approaches. A separate held-out generalization test is a natural next step, which we are pursuing through an ongoing expansion using MIMIC-IV patient data [9] and clinician-sourced questions.

**Single baseline.** The evaluation compares MedSQLX against a single baseline (DIN-SQL). While DIN-SQL represents a strong prompting-based approach using the same underlying model, additional baselines and architectural ablations would provide a more comprehensive assessment of the relative contribution of each system component. We leave this to future work alongside the benchmark expansion.

**DuckDB dependency.** The current implementation targets DuckDB specifically, leveraging its native support for SQL macros and Python UDF registration. Extending the framework to other database systems would require adapting the DuckDB Executor Tool and UDF Validator Tool to the target system's UDF registration API and type system. The Semantic Analysis Layer and agent coordination logic contain no database-specific assumptions and would not require modification.

## 4 CONCLUSION

We are the first to present an agent-based framework called MED-SQLX for generating medical UDFs from natural language. MED-SQLX combines a chain-of-clarifications reasoning method for autonomous underspecification resolution and the AgentFlow reasoning framework for code generation. It achieves 100% accuracy across all ten evaluation scenarios, outperforming DIN-SQL, a state-of-the-art NL-to-SQL system, by 2× on accuracy. It demonstrates

the effectiveness of structured reasoning and tool orchestration, combining domain knowledge retrieval, live database execution, and LLM-based validation in a self-correcting loop.

In the future, we will extend MEDSQLX along several important directions. First, we will expand the framework to support richer data modalities, such as medical imaging and clinical notes, by integrating multi-modal UDFs and cross-modal reasoning. Second, we aim to improve the robustness and generalizability of UDF generation by incorporating more medical knowledge bases, such as our prior works on systematic review systems [10], and larger-scale evaluation on diverse real-world datasets beyond MedAgentBench. Finally, we will explore tighter integration with our prior works on UDF-centric query optimizers [27–29] to bridge the gap between logical correctness and system-level efficiency. We will also strengthen the evaluation by using more baseline NL-to-SQL systems, and more types of LLM models.

## AUTHORS

**Catlynh Nguyen** is an undergraduate student at Arizona State University pursuing dual degrees in Computer Science and Mathematics. *Community: Data Management.*

**Irbaz Bin Riaz (M.D., and Ph.D.)** is an oncologist, and Assistant Professor at Mayo Clinic-Arizona, where he leads AI-assisted cancer research. *Community: Biomedical.*

**Chitta Baral (Ph.D.)** is a Professor in the School of Computing and Augmented Intelligence at Arizona State University. *Community: AI and Natural Language Processing.*

**Jia Zou (Ph.D.)** is an Assistant Professor in the School of Computing and Augmented Intelligence at Arizona State University. *Community: Data Management.*

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
