# OpenReview forum: "MedSQLX: Translation of Medical Queries into UDF-Centric SQL"
_VLDB.org/2026/Workshop/BioDMS — BioDMS 2026 ProjectTalk_

### Official Review · Reviewer_zFS2 · 2026-06-08

**Summary:**

Dear Authors,

I have reviewed your manuscript entitled “MedSQLX: Translation of Medical Queries into UDF-Centric SQL.” The goal of the paper is to address the problem of translating natural-language clinical task descriptions in the medical domain into SQL queries that require user-defined functions (UDFs). The authors propose a novel framework called “MedSQLX”. Specifically, MedSQLX is a two-layer agentic framework consisting of a semantic analysis layer and a code generation layer. The framework is evaluated on ten medical query scenarios and 57 test cases, covering tasks such as disease staging, risk scoring, trend analysis, and temporal reasoning. The reported results show that MedSQLX achieves perfect performance on all evaluated test cases and substantially outperforms a DIN-SQL baseline, highlighting the potential benefits of implementing their novel framework.

**Confidence Of Review:**

4

**Detailed Feedback Points:**

1) The paper addresses an interesting and relevant problem, that is, the generation of SQL queries that rely on domain-specific user-defined functions (UDFs). The focus on medical applications further enhances the practical relevance of the work.

2) The proposed framework is clearly described and the overall workflow is easy to follow.

3) The results obtained from their preliminary evaluation are promising and demonstrate substantial improvements over the DIN-SQL baseline across the evaluated scenarios.

4) While the architecture is clearly presented, the rationale behind several design choices remains unclear. For example, there is not justification about why they chose a two-layer architecture, nor does it discuss alternative designs or explain why this decomposition is expected to be particularly effective for the problem at hand. The paper would definitely be strengthened if the authors provided additional justification for the main architectural choices.

5) The comparison is performed against a single baseline, which does not seem to work at all in most of the scenarios. Either including additional existing approaches or variations of their MedSQLX ( to determine the best architecture for their novel framework) would provide a more comprehensive assessment of the relative strengths and limitations of the proposed framework.

**Relevance For Biodms:**

3

---

### Official Review · Reviewer_X52s · 2026-06-10

**Summary:**

This paper presents an interdisciplinary project combining biomedical and data management research, focusing on translating natural language medical queries into UDF-centric SQL. The work introduces a method (MedSQLX) and evaluates it on a medical dataset, demonstrating improvements over a prior approach.

**Confidence Of Review:**

3

**Detailed Feedback Points:**

- Relevance and "BioDMS" Potential: The combination of authors and the chosen topic are clearly well aligned with the BioDMS workshop goals. The work is at the intersection of biomedical data and data management systems, and the focus on translating medical queries into SQL has strong potential to enable new capabilities in clinical data analysis. Overall, the paper presents a promising interdisciplinary direction that could lead to impactful future work in both communities.

- Clarity of Biomedical Motivation and Scoping: It is somewhat difficult to understand the true relevance to the biomedical domain early in the paper. While the medical setting is used throughout and reflected in the experimental data, the paper would benefit from a clearer articulation of what challenges are unique to the biomedical domain in the context of UDF-centric SQL. For example, what characteristics of medical data or clinical workflows make this problem particularly compelling or different from other domains? One way to strengthen this would be to introduce a concrete, end-to-end example earlier in the paper (potentially based on the experimental data). Some useful examples do appear later (e.g., early in Section 2.1), but presenting and expanding them earlier would help better ground the contribution.

- Evaluation Scope and Benchmarking: The experimental setup could be further strengthened in future iterations. While the paper references a large dataset (785,207 medical records), the evaluation is ultimately conducted on a relatively small number of test cases (57 queries). In addition, the method achieves perfect performance on this subset, which raises questions about the difficulty and representativeness of the benchmark. Expanding the evaluation to include more diverse and challenging test cases would improve confidence in the generalizability of the approach. That said, the method does outperform a state-of-the-art baseline, and for the scope of a workshop paper, the current evaluation is sufficient to demonstrate promise.

**Relevance For Biodms:**

3

---

### Official Review · Reviewer_ZzFd · 2026-06-11

**Summary:**

The authors suggest and evaluate an agentic LLM framework, MedSQLX, that employs multiple agents with different tasks, also for validation, to map natural language queries by non-expert users in a clinical context to partially complex user-defined function SQL queries in DuckDB.

**Confidence Of Review:**

1

**Detailed Feedback Points:**

Strength
- The performance is of course great (see some slight reservations below), as is the comparison to another NL-to-SQL pipeline; more methods would be better but has to remain part of future work.

Weaknesses
- It does not become clear on which tasks the pipeline was developed. Distinguishing this from the 57 tasks it was later evaluated on would be highly important, especially to assess whether the perfect precision that is observed is genuine. Along the same lines, why were not 6 test cases per task defined but sometimes only 5? This looks a bit arbitrary, especially given that their tool MedSQLX scores perfectly.
-	How strongly is the approach limited to DuckDB – can it be extended, what is needed for that?

Further points

A more detailed evaluation on specific tasks completion would be of interest, e.g. the underspecification detector – how good is it at finding those tasks; but it is out of scope here.
It would be also interesting to discuss data security/privacy issues: how much do the agents/LLM need to know about the data they are querying and/or its structure?

**Relevance For Biodms:**

4